# A Risk Assessment System of Toxic Gas Leakages in Metallurgy Based on Fuzzy Evaluation with Comprehensive Weighting

Kaeryaer Kariman [ID], Na Gao *[ID], Chunguo Ma and Zhao Wang

Key Laboratory of Ministry of Education for High Efficient Mining and Safety, University of Science and Technology Beijing, Beijing 100083, China; s20200120@xs.ustb.edu.cn (K.K.); g20198140@xs.ustb.edu.cn (C.M.); s20200129@xs.ustb.edu.cn (Z.W.)
* Correspondence: aggie198308@ustb.edu.cn; Tel.: +86-138-1027-8308

**Abstract:** Toxic gas leakage in metallurgic plants has emerged with the growth of crude steel production in recent years, causing damage to people, facilities, and the environment. Poisonous gas leakage can lead to other severe accidents including fires, explosions and gas poisoning. In this paper, we propose a risk assessment system (RAS) for toxic gas leakage using a fuzzy evaluation method integrating the entropy weighting method (EWM) and the order relationship method (ORM) and compiled an index system consisting of four first-level indices and fifteen secondary indices. The first-level indices are blast furnace safety performance, protective facilities, evacuation and dilution facilities, and poisonous gas management. The four first-level indices' toxic gas leak evaluation result is 0.8581, 0.8971, 0.7733, and 0.8652, respectively. We observe that the overall status of the metallurgical plant is "excellent", yet the result for the evacuation and dilution facilities was less than 0.8, indicating that there is still room for improvement. The risk evaluation time is reduced by forty percent by adopting RAS.

**Keywords:** gas leakage; metallurgy; fuzzy comprehensive evaluation; risk assessment





## 1. Introduction

Metal smelting enterprises involve complex production processes that pose various risks [1,2]. Toxic gas leakage is one of the most serious accidents in steel plants, which may lead to secondary accidents, including fires, explosions, and gas poisoning. The crude steel production in many countries has increased to varying degrees recently [3]. Simultaneously, the risk of toxic gas leaks in steel plants increases with production growth [4–6].

Quantitative risk analysis (QRA) is widely applied in estimating the risk of an accident and the severity of the incident [7,8]. Lee [9] has constructed a fuzzy membership function based on the closest point of approach (CPA) and the time to CPA (TCPA) in order to recognize dangerous situations in vessel traffic surveillance. Wyszynski [10] has adopted an analytical hierarchy process (AHP) to manage crisis and support the decision-making process. Tanoli [11] has improved the rockfall risk assessment system for Pakistan by quantifying animal activity along the highways. The entropy weighting method (EWM) is one of the weighting methods adopted in the fuzzy methods [12]. Xu et al. [13] have adopted the improved entropy weight method to evaluate the risk of urban floods, while Liang et al. [14] have used the entropy method to quantitatively analyze the uncertainty in the process of weight calculation. The order relationship method (ORM) is used as a tool in assessing and evaluating risk levels [15]. Chen et al. [16] have applied the ORM to calculate indicator weights in order to evaluate the city innovation capability in Liaoning province, China. Jiang [17] has simulated the gas concentration changes at various points in order to guide emergency rescues after gas leaks. Seo et al. [18] have developed a system based on QRA to suggest an optimal evacuation route in the circumstances of gas leakage.

However, the application of the fuzzy evaluation method in the risk assessment of toxic gas leaks in steel manufacturing plants to prevent accidents is rare, which makes

the existing risk of such gas leakage accidents challenging to detect, and the identification of risk factors is not comprehensive. The risk analysis time is long in the existing risk assessment approach, and the leakage accident cannot be warned in time and effectively. Consequently, it is necessary to carry out a comprehensive risk factor analysis of gas leakage in the metal smelting process and adopt principles of fuzzy mathematics to determine and calculate the importance of each risk factor based on the various complex causes of gas leaks in metal smelting enterprises.

The objective of the present research is to evaluate the risk of toxic gas leakage in a steel plant in order to promote the efficiency of gas leak risk assessment. We propose a comprehensive weighting method (CWM), integrating the entropy weighting method (EWM) and the order relationship method (ORM) to calculate the weight vectors (Section 2). By analyzing toxic gas leak accidents and related literature surveys [19,20], the evaluation indexes are confirmed using the Delphi method [21] and the clustering method [22], and a toxic gas leakage index system is established based on these evaluation indexes (Section 3). A risk assessment system (RAS) for toxic gas leakage is developed (Section 3). The system iss applied to evaluate each factor's risk level and importance ranking, and the results prove effective (Section 4). RAS enriches the risk factors of the existing safety checklist and sorts the importance of the indexes, which can optimize the distribution of management resources. The time cost of a gas leakage risk assessment is reduced after applying RAS (Section 5).

## 2. Methods

This section introduces the methods of compiling an evaluation index system and evaluating the gas leak risk. Meanwhile, a comprehensive weighting method is proposed by integrating EWM and ORM.

### 2.1. Determination of Evaluation Index System

We investigated gas leakage accidents occurring over the past twenty years in metallurgical plants and classified the accidents according to the production process. As toxic gas leakage accidents mainly occur during the iron-making process, we focused on the influencing factors of gas leakage accidents in the iron-making process when determining the evaluation indices. The construction of the index system is a holistic process, from the whole to the part. It should start with consideration of the overall situation, followed by analysis of the characteristics of decision-making goals and their influencing factors, grasping the relationships between indicators, and selecting scientific and reasonable indicators. The specific process is shown in Figure 1.

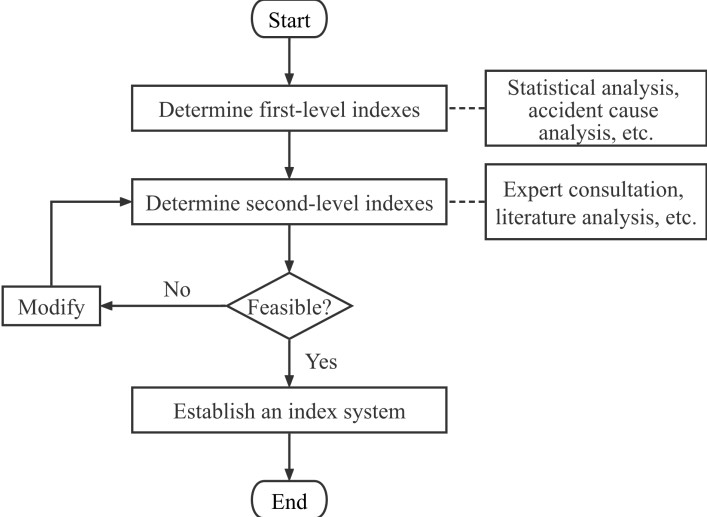

**Figure 1.** Index determination process.

### 2.2. Construction of the Weight Sets for Evaluation Indices

As iron and steel smelting combines human operations and programmatic automation in the metallurgical industry, purely subjective or objective weighting analysis is not sufficient [23]. Consequently, the comprehensive weighting method (CWM) is proposed, which integrates the entropy weighting method (EWM) and the order relationship method (ORM) to determine the weights for the risk analysis of toxic gas leakage accidents.

#### 2.2.1. Entropy Weighting Method

In this paper, the EWM is used as an objective method in the calculation that can help to reduce the subtle deviation of subjective scoring of on-site workers in the steel plant and avoid the deviation caused by human factors. The EWM uses on-site expert scoring in the calculation. Experts rate each indicator from 0 to 100, with a score of 90 to 100 as excellent, a score of 80–89 as good, a score of 70–79 as moderate, and a score of 60–69 classified as defective; scores below 60 are classified as poor. Accordingly, we can obtain the value of $r_i$ by summing the times that the same judgment is made of the index, then dividing by the number of people considered, and the entropy value $e_i$ of each indicator is calculated using the value of $r_{ij}$ [12].

$$e_i = -\frac{1}{ln(n)} \sum_{j=1}^{n} \frac{r_{ij}}{\sum_{j=1}^{n} r_{ij}} ln\left(\frac{r_{ij}}{\sum_{j=1}^{n} r_{ij}}\right), (i = 1, 2, \ldots, m),$$ (1)

Finally, the weight vector $a_i$ can be obtained [13]. The weight vector A is composed of $a_i$, calculated by EWM:

$$a_i = \frac{1 - e_i}{\sum_{i=1}^{n}(1 - e_i)}, (0 \le a_i \le 1).$$ (2)

#### 2.2.2. Order Relationship Method

The order relationship method could reflect the degree of importance of different secondary indices in the same first-level index. The ORM requires a pre-determined order relationship for each indicator, according to the evaluation index; namely, setting a sequence for the evaluation index of the gas leak. Then, the relative importance of each adjacent evaluation index is judged; that is, we determine the degree of importance based on the ratio of the expert's scoring of the two indices. This step serves to calculate the degree of importance between two adjacent indices, as shown in Equation (5) [15]:

$$x_i = \frac{u_i}{u_{i-1}}(i = n - 1, n - 2, \ldots, 2),$$ (3)

where $x_i$ is the ratio of the expert's scoring between two indices, and $u_i$ is the evaluation index. The value of $x_i$ varies from 1 to 1.8. When the value of $x_i$ is 1, it represents that the adjacent evaluation indices are equally important; meanwhile, when the value of $x_i$ is 1.8, it represents that the current index is much more important than the previous one. The weight of the index can be calculated based on the $x_i$, as shown in the equation below [16].

$$b_n = (1 + \sum_{i=2}^{n} \prod_{i=2}^{n} x_i)^{-1},$$ (4)

$$b_{i-1} = x_i b_i, (i = n, n - 1, \ldots, 3, 2),$$ (5)

where $b_i$ is the element of the weight vector B, calculated using the ORM.

#### 2.2.3. Comprehensive Weighting Method

The CWM combines both the EWM and ORM as a combination of subjective and objective weighting methods, allowing us to obtain the comprehensive weights of the evaluation indices for toxic gas leakage. The steps of construction of the weight sets for

an evaluation index are shown in Figure 2. According to the principle of addition and multiplication integrations of the meta-synthesis [24], the following two index weights are obtained.

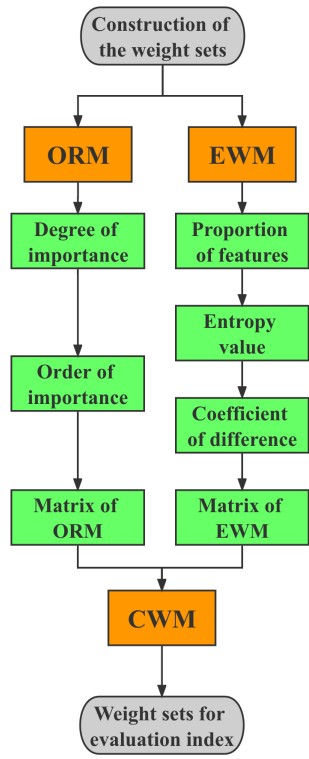

**Figure 2.** Steps of construction for the weight sets.

$$w_i = \frac{a_i b_i}{\sum_{i=1}^{n} a_i b_i}, i = 1, 2, \dots, n, \quad (6)$$

$$w_i = k_1 a_i + k_2 b_i, i = 1, 2, \dots, n, \quad (7)$$

$$k_1 = \frac{\sum_{i=1}^{n} \sum_{j=1}^{m} a_i r_{ij}}{\sqrt{\left(\sum_{i=1}^{n} \sum_{j=1}^{m} a_i r_{ij}\right)^2 + \left(\sum_{i=1}^{n} \sum_{j=1}^{m} b_i r_{ij}\right)^2}}, \quad (8)$$

$$k_2 = \frac{\sum_{i=1}^{n} \sum_{j=1}^{m} b_i r_{ij}}{\sqrt{\left(\sum_{i=1}^{n} \sum_{j=1}^{m} a_i r_{ij}\right)^2 + \left(\sum_{i=1}^{n} \sum_{j=1}^{m} b_i r_{ij}\right)^2}}, \quad (9)$$

where $a_i$ represents the result of the EWM for element $i$; $b_i$ represents the result of the ORM for element $i$; and $k_1$ and $k_2$ are the EWM and ORM weight coefficients, respectively. $k_1$ and $k_2$ can be determined with respect to the requirements and conditions of different situations.

### 2.3. Fuzzy Comprehensive Evaluation

Fuzzy comprehensive evaluation is obtained based on the evaluation index system of poisonous gas leakage and its weights, where the fuzzy comprehensive evaluation involves confirming the fuzzy parameter set, fuzzy membership matrix, and fuzzy comprehensive evaluation vector. The fuzzy parameter set often comprises a two-level fuzzy evaluation index system. The first-level evaluation index set $V$ includes the overall influencing factors of toxic gas leakage:

$$V = \{V_1, V_2, \dots V_m\}, \quad (10)$$

where $V$ is the fuzzy comprehensive evaluation index system, $V_m$ is the first-level index, and $m$ is the number of first-level parameters. The second-level evaluation indicator set $u_{ij}$ is a specific index of the critical factors of the first-level index:

$$
\begin{aligned}
V_1 &= \{u_1, u_2, \ldots, u_n\}, \\
V_2 &= \{u_{n+1}, u_{n+2}, \ldots, u_{n+c}\},
\end{aligned}
\tag{11}
$$

where $u_i$ is the second-level index, and $i$ is the number of the second-level parameters. As $u_i$ is continuous, the number of $u_i$ in $V_2$ is the continuation of $V_1$.

Then, the evaluation set for the indices can be established. Experts and workers on site have different degrees of judgment with respect to the safety status of the secondary-level indices. According to the actual situation of the safety devices, five degrees of safety judgment are usually adopted. The evaluation set is denoted as $S = \{S_1, S_2, S_3, S_4, S_5\}$, where $S_1$ denotes that the safety status of toxic gas leakage is excellent, $S_2$ is good, $S_3$ is moderate, $S_4$ is defective, and $S_5$ is poor safety.

Subsequently, the fuzzy membership matrix $R$ includes the statistical value of the judgment of the second-level index $u_i$. Each individual matrix $R$ corresponds to the evaluation value of an individual first-level index of toxic gas leak risk. The statistical data $r_{ij}$ indicate the different judgment results for the second-level indices.

$$
R = \begin{bmatrix}
r_{11} & r_{12} & \cdots & r_{1n} \\
r_{21} & r_{22} & \cdots & r_{2n} \\
\vdots & \vdots & \ddots & \vdots \\
r_{m1} & r_{m2} & \cdots & r_{mn}
\end{bmatrix}.
\tag{12}
$$

After the fuzzy membership matrix $R$ is obtained, the evaluation vector $F$ can be obtained by multiplying the weight vector $w$ and fuzzy membership matrix $R$, as shown in the Equation below:

$$
F = w \times R.
\tag{13}
$$

The result of the elements of evaluation vector F represents the safety level. A result of $f_1$ represents an "excellent" score for the risk of toxic gas leakage, $f_2$ represents a "good" score, $f_3$ represents a "moderate" score, $f_4$ represents a "defective" score, and $f_5$ represents a "poor" score. Finally, the maximum value of $f_i$ represents the ultimate evaluation result.

*2.4. Risk Assessment System Feedback*

The proposed RAS adopts a closed-loop management model to effectively control and promptly feed back the potential risk of a toxic gas leak in a metal smelting factory. Therefore, the methods mentioned above are used with the aim of obtaining a relatively precise evaluation result and to confirm the rationality of the RAS index system to improve the risk factor conditions. The framework of the closed-loop method for the risk assessment system is shown in Figure 3.

The closed-loop method for the RAS can be divided into two parts: one is the RAS proposal, and the other is the RAS verification.

- The RAS assesses toxic gas leakage risk through the index system in order to provide feedback on the current risk status.
- The risk of toxic gas leakage evaluated by the RAS is checked to validate the suitability of the index system in order to promote the performance of the RAS in evaluating the toxic gas leakage risk.

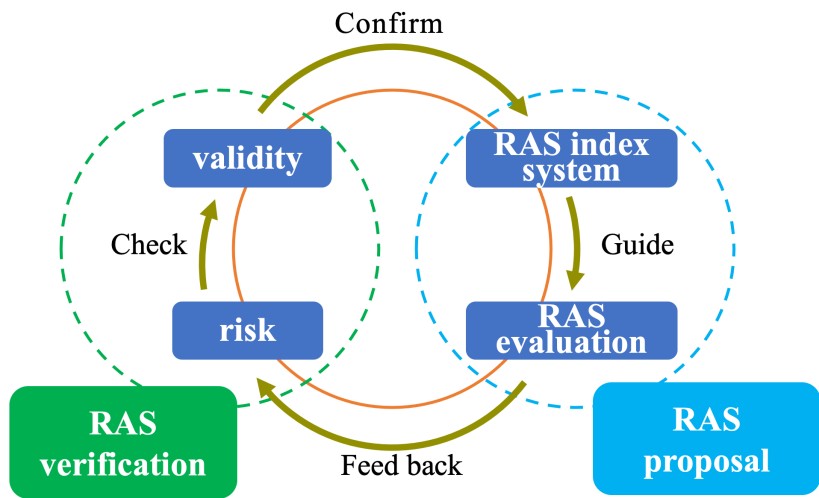

**Figure 3.** Closed-loop framework of the risk assessment system.

## 3. Risk Assessment System Development

In this section, the risk assessment system (RAS) was developed based on the methods introduced in Section 2. For the proposed system, we constructed an architecture design based on the risk analysis of poisonous gas leakage accidents in metal smelting enterprises in order to realize the intelligent risk analysis of poisonous gas leakages in steel plants through the use of scientific and reasonable methods.

### 3.1. Logical Architecture

We selected the B/S structure as the logical structure for the system (i.e., the server plus browser structure), where the experts and on-site workers only need to use a device with a browser that is connected to the Internet, without installing other software. At the logical level, the client-side Web application server and browser structure were selected for this system. The interaction with the user is mainly dominated by the client web browser, which sends a request to the server after a user's operation and then accepts and analyzes the data returned by the server to display it to the user. It mainly receives user requests, forms SQL statements on the basis of parsing the request parameters, and then sends operation commands (e.g., queries, modifications, additions, and deletions) to the database for execution. Finally, it returns the data to the client. The logical structure of the system is shown in Figure 4.

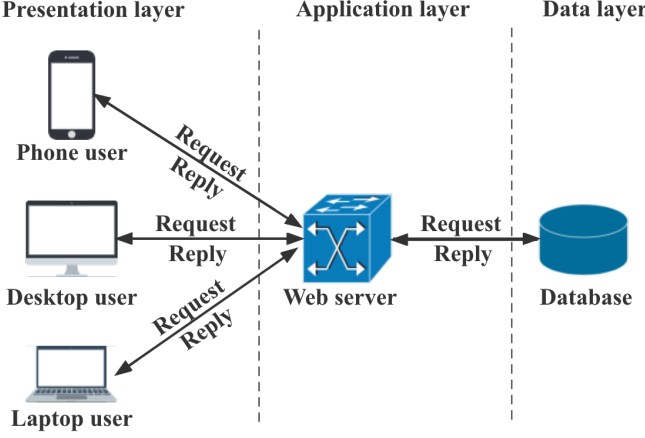

**Figure 4.** Logical structure of the risk assessment system.

*3.2. Technical Framework*

The system adopts a layered design, which is divided into four layers: the data access layer, logic layer, control layer, and the data presentation layer. The overall technical framework is shown in Figure 5.

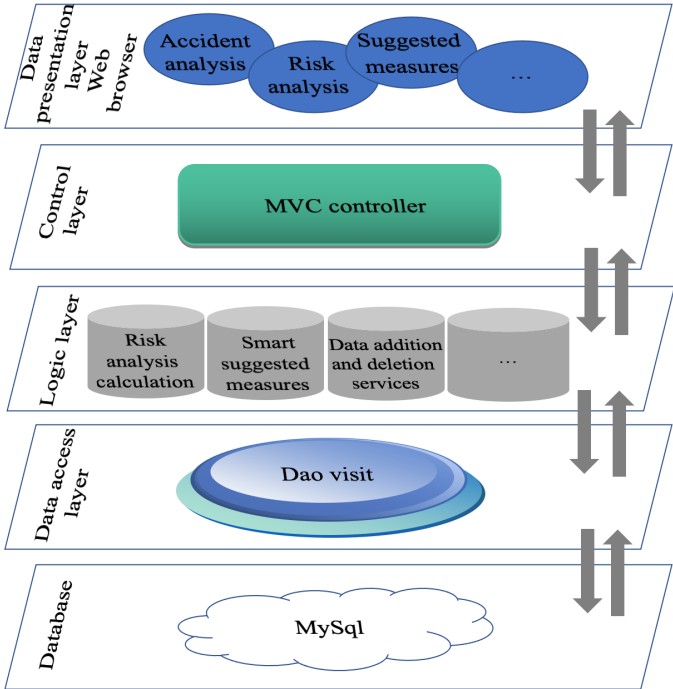

**Figure 5.** Technical framework of the risk assessment system.

The presentation layer is intended for the experts and on-site workers in the steel plant and is responsible for processing the user interface, collecting user data, and displaying data to the user. The user interface is the frontend of a Web application, and generally uses HTML and CSS for page layout, as well as JavaScript to achieve page data processing. In order to improve development efficiency, we used a jQuery-based Easy UI plug-in for development and Ajax technology to achieve partial refreshing of the page.

In both the business and presentation layers, the control layer acts as a bridge and plays the function of forwarding requests in the system. The control layer is responsible for intercepting the user requests, encapsulating the user data, and passing it to the business logic layer. The business logic layer then performs corresponding processing, according to the user's request, and returns the result to the client. The entire request processing uses the MVC model for control.

The business logic layer is the core of the Web application, enabling business logic to be realized. The requested data is sent through the control layer, received by the business logic layer, and then corresponding logic processing is performed to return the result to the control layer. It can also provide manipulated data for the data layer, as well as receive the result returned by the data layer.

Realizing interaction with the database is the core function of the data access layer. We selected Tomcat for database management. The data layer provides basic creation, query, and deletion operations for data objects, which correspond to associated operations in the database, such as queries, modifications, deletions, and additions. These operations enable users to continue accessing the database and ensure access to related services through the business logic layer.

As introduced above, in the calculation of the fuzzy comprehensive evaluation, the implementation process in RAS is summarized in Algorithm 1:

---

**Algorithm 1** General process of fuzzy comprehensive evaluation in the risk assessment system

---

**Input:** statistical vector X= $(X_1, X_2, \ldots, X_n)$ of all the indices
**Output:** evaluation vector F
  Build the fuzzy membership matrix R
  **for** each $i \in [1, n]$ **do**
    **for** each $j \in [1, m]$ **do**
      $r_{ij} = p_{ij} / \sum_{j=1}^{m} p_{ij}$
    **end for**
  **end for**
  calculate the weight vector A of EWM
  Build the ratio matrix X
  **for** each $i \in [1, n]$ **do**
    $x_i = u_i / u_{i-1}$
  **end for**
  calculate the weight vector B of ORM
  calculate the comprehensive weight vector W of CWM
  calculate the comprehensive evaluation vector F

---

*3.3. Basic Process of the Risk Assessment System*

    The RAS of toxic gas leakages in metallurgy was developed based on online processing and computational calculations. The key steps are depicted in Figure 6.

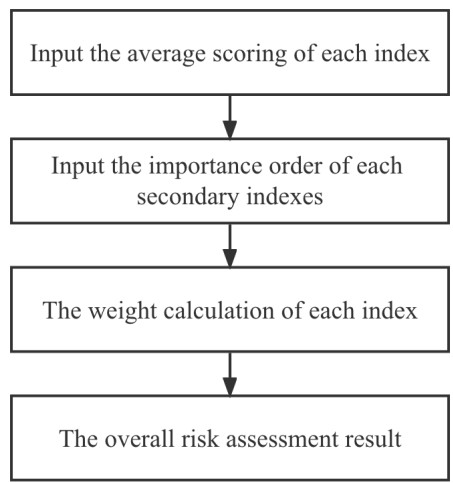

**Figure 6.** Basic process of the proposed risk assessment system.

(1)   Input the average score of each index of the RAS for metallurgic toxic gas leakages, which was previously checked by on-site workers and experts.
(2)   Input the importance order of each secondary index in the first-level indices, which on-site workers and experts previously checked.
(3)   Calculation of RAS is the third step. When the necessary figures have been obtained, the weight of each index can be processed and calculated efficiently using the method mentioned in Section 2.
(4)   The overall risk assessment result is shown eventually, and the safety status of each first-level index can be obtained.

    The calculation result is the evaluation vector $F$, which indicates the actual probability of each assessment status, and the RAS result is the assessment status with the highest probability.

*3.4. Risk Assessment System Index System Construction*

According to the existing metal smelting industry norms [25], national standards [26–28], relevant literature research results [29–32], and a large number of statistical data [33], we conducted a risk analysis and regular summary of toxic gas leakage accidents in the metal smelting process to establish a hierarchical evaluation system. The evaluation objective of this index system is the risk level of toxic gas leakage during metal smelting processes, including two levels of indicators. The specific indicators are described in the following. The first-level index set for RAS can be described as V = $\{V_1, V_2, V_3, V_4\}$ = {blast furnace safety performance, protective facilities, evacuation and dilution facilities, and the aging degree of the gas tank}. In addition, the set of second layer indices can be described as follows: $V_1 = \{u_{11}, u_{12}, u_{13}, u_{14}\}$ = {three-way valve effectiveness, gas water seal height, corrosion degree of gas tank, and the aging degree of the gas tank}, $V_2 = \{u_{21}, u_{22}, u_{23}, u_{24}\}$ = {CO monitoring alarm device, mechanical ventilation device, CO concentration detector, and the low pressure alarm means}, $V_3 = \{u_{31}, u_{32}, u_{33}, u_{34}\}$ = {emergency evacuation route, evacuation channel settings, DCS system, and the nitrogen sealing device}, $V_4 = \{u_{41}, u_{42}, u_{43}\}$ = {management system, monitoring duty, and the emergency plan}. The specific hierarchical structure is shown in Table 1.

**Table 1.** Risk assessment index system.

| First-Level Indices | Number | Secondary Indices | Number |
|---|---|---|---|
| Blast furnace safety performance | $V_1$ | Three-way valve effectiveness | $u_{11}$ |
| | | Gas water seal height | $u_{12}$ |
| | | Corrosion degree of gas tank | $u_{13}$ |
| | | Aging degree of gas tank | $u_{14}$ |
| Protective facility | $V_2$ | CO monitoring alarm device | $u_{21}$ |
| | | Mechanical ventilation device | $u_{22}$ |
| | | CO concentration detector | $u_{23}$ |
| | | Low pressure alarm means | $u_{24}$ |
| Evacuation and dilution facilities | $V_3$ | Emergency evacuation route | $u_{31}$ |
| | | Evacuation channel settings | $u_{32}$ |
| | | DCS system | $u_{33}$ |
| | | Nitrogen sealing device | $u_{34}$ |
| Poisonous gas management | $V_4$ | Management System | $u_{41}$ |
| | | Monitoring duty | $u_{42}$ |
| | | Emergency plan | $u_{14}$ |

## 4. Results and Discussion

In this section, we present the results of an evaluation in a metallurgy steel plant, which was used as an example to obtain an objective result regarding the plant's risk status of toxic gas leak accidents based on the methods in Section 2 and the system developed in Section 3.

*4.1. Risk Assessment Results of Toxic Gas Leak*

This section combines the methods mentioned in Section 2 to calculate the weight vector based on both the degree of dispersion of the indices as well as the importance order. Thus, we need to obtain the appraisal and importance relationship of each secondary index through the Delphi and clustering methods. We invited ten iron- and steel-producing experts and integrated their scores. The fuzzy membership matrix for the toxic gas leak risk is shown in Table 2.

The value of $r_{ij}$ represents the proportion of judgment for a second-level index, and the whole $r_{ij}$ of a first-level index form a matrix R. Therefore, the matrix R for each first-level index is different. The matrix $R_1$ is shown below, and $R_2$, $R_3$, and $R_4$ can be obtained similarly.

$$R_1 = \begin{bmatrix} 0.8 & 0.1 & 0.1 & 0 & 0 \\ 0.7 & 0.2 & 0.1 & 0 & 0 \\ 1 & 0 & 0 & 0 & 0 \\ 0.9 & 0 & 0.1 & 0 & 0 \end{bmatrix}.$$

**Table 2.** Fuzzy membership matrix of toxic gas leak risk.

| First-Level Indices $V_i$ | Secondary Indices $u_{ij}$ | Membership $r_{ij}$ | | | | |
|---|---|---|---|---|---|---|
| | | Excellent | Good | Moderate | Defective | Poor |
| Blast furnace safety performance | Three-way valve effectiveness | 0.8 | 0.1 | 0.1 | 0 | 0 |
| | Gas water seal height | 0.7 | 0.2 | 0.1 | 0 | 0 |
| | Corrosion degree of gas tank | 1 | 0 | 0 | 0 | 0 |
| | Aging degree of gas tank | 0.9 | 0 | 0.1 | 0 | 0 |
| Protective facility | CO monitoring alarm device | 1 | 0 | 0 | 0 | 0 |
| | Mechanical ventilation device | 0.8 | 0.1 | 0.1 | 0 | 0 |
| | CO concentration detector | 0.8 | 0.2 | 0 | 0 | 0 |
| | Low pressure alarm means | 0.9 | 0 | 0.1 | 0 | 0 |
| Evacuation and dilution facilities | Emergency evacuation route | 0.7 | 0.1 | 0.2 | 0 | 0 |
| | Evacuation channel settings | 0.7 | 0.2 | 0.1 | 0 | 0 |
| | DCS system | 0.8 | 0.1 | 0.1 | 0 | 0 |
| | Nitrogen sealing device | 0.9 | 0.1 | 0 | 0 | 0 |
| Poisonous gas management | Management System | 0.8 | 0.2 | 0 | 0 | 0 |
| | Monitoring duty | 0.9 | 0.1 | 0 | 0 | 0 |
| | Emergency plan | 0.9 | 0 | 0.1 | 0 | 0 |

The importance gradation was used in the calculation of ORM, and the specific values are shown in Table 3. The importance order in Table 3 shows the rank of the secondary indices in the first-level index, where the importance ratio $x_{ij}$ represents the importance degree of an indicator to the next important indicator. The ranks and the importance ratios are obtained through experienced experts in the steel plant.

**Table 3.** Importance gradation of the toxic gas leak risk.

| First-Level Indices | Secondary Indices | Importance Order | Importance Ratio $x_{ij}$ |
|---|---|---|---|
| Blast furnace safety performance | Three-way valve effectiveness | 4 | - |
| | Gas water seal height | 3 | 1.2 |
| | Corrosion degree of gas tank | 1 | 1.2 |
| | Aging degree of gas tank | 2 | 1.4 |
| Protective facility | CO monitoring alarm device | 1 | 1.2 |
| | Mechanical ventilation device | 2 | 1.0 |
| | CO concentration detector | 4 | - |
| | Low pressure alarm means | 3 | 1.2 |
| Evacuation and dilution facilities | Emergency evacuation route | 4 | - |
| | Evacuation channel settings | 3 | 1.4 |
| | DCS system | 2 | 1.2 |
| | Nitrogen sealing device | 1 | 1.0 |
| Poisonous gas management | Management System | 1 | 1.0 |
| | Monitoring duty | 2 | 1.0 |
| | Emergency plan | 3 | - |

Based on the original data of Tables 2 and 3, the comprehensive weights of the first-level indices were obtained using Equations (1)–(9).

Based on the evaluation results, the weight vectors of the first-level indicators are presented in Table 4. The weight set $w_i$ was calculated for every second-level index, where the weights of the second-level indices in the same first-level index form a weight vector $W_i$.

By using the weight information, the risk of a toxic gas leak can be evaluated through the fuzzy comprehensive evaluation method. The evaluation index weight $w_i$ and the fuzzy membership matrix $R_i$ were substituted into the evaluation method of the RAS in Section 2.3.

**Table 4.** Weight vectors of first-level indices.

| First-Level Indices | Number | Weights of Secondary-Level Indices | | | |
|---|---|---|---|---|---|
| | | $w_1$ | $w_2$ | $w_3$ | $w_4$ |
| Blast furnace safety performance | $W_1$ | 0.2996 | 0.2078 | 0.2958 | 0.1967 |
| Protective facility | $W_2$ | 0.3801 | 0.1910 | 0.2183 | 0.2106 |
| Evacuation and dilution facilities | $W_3$ | 0.2533 | 0.2533 | 0.2536 | 0.2398 |
| Poisonous gas management | $W_4$ | 0.3477 | 0.2496 | 0.4027 | - |

Then, the evaluation vector was obtained. First, it was necessary to calculate the membership matrix based on Table 3 before obtaining the evaluation vector. Taking the first-level index $V_1$ (blast furnace safety performance) as an example, the weight vector of $W_1$ was (0.2996, 0.2078, 0.2958, 0.1967). Based on the fuzzy membership matrix $R_1$ and the weights of $V_1$, the evaluation value of $F_1$ was calculated using Equation (13):

$$F_1 = W_1 \times R_1 = \begin{bmatrix} 0.2996 \\ 0.2078 \\ 0.2958 \\ 0.1967 \end{bmatrix}^T \times \begin{bmatrix} 0.8 & 0.1 & 0.1 & 0 & 0 \\ 0.7 & 0.2 & 0.1 & 0 & 0 \\ 1 & 0 & 0 & 0 & 0 \\ 0.9 & 0 & 0.1 & 0 & 0 \end{bmatrix} = \begin{bmatrix} 0.8581 \\ 0.0715 \\ 0.0704 \\ 0 \\ 0 \end{bmatrix}^T .$$

Similarly, the first-level comprehensive evaluation vectors for other indices were obtained. The evaluation vectors for all indices are provided in Table 5.

Based on the principle of maximum membership, $F_1 = \max \{f_i\} = 0.8581$, and thus, the blast furnace safety performance of the steel plant was found to be "excellent," being the highest in rank [34]. Similarly, the design levels of the RAS in other branches of the index were also evaluated, and the evaluation results are shown in Table 6.

### 4.2. Weight Analysis of Risk Assessment System

To verify the accuracy and superiority of the CWM, the weights obtained by ORM, EWM, and CWM for the same second-level index are presented in Figure 7. It can be observed that the weights of ORM tended to be related to the importance order—the more important the index, the bigger its weight. In poisonous gas management, the importance ratio for both indices are 1, which means that they are equally important; therefore, there is no difference in importance of these first-level indices, which could explain why the ORM weights in Figure 7d do not completely match the importance order. Furthermore, the weights obtained by EWM can be seen to under-estimate the risk of the first index and over-estimate the last index, which may lead to the ignorance of the first index's risk or exaggeration of the last index's risk.

**Table 5.** The value of evaluation sets of toxic gas leakages.

| First-Level Indices $V_i$ | Number | Evaluation Set | | | | | Max $\{f_i\}$ | Result |
|---|---|---|---|---|---|---|---|---|
| | | Excellent $f_1$ | Good $f_2$ | Moderate $f_3$ | Defective $f_4$ | Poor $f_5$ | | |
| Blast furnace safety performance | $F_1$ | 0.8581 | 0.0715 | 0.0704 | 0 | 0 | $f_1$ | excellent |
| Protective facility | $F_2$ | 0.8971 | 0.0628 | 0.0402 | 0 | 0 | $f_1$ | excellent |
| Evacuation and dilution facilities | $F_3$ | 0.7733 | 0.1253 | 0.1014 | 0 | 0 | $f_1$ | excellent |
| Poisonous gas management | $F_4$ | 0.8652 | 0.0945 | 0.0403 | 0 | 0 | $f_1$ | excellent |

**Table 6.** Evaluation value of the safety checklist method.

| First-Level Indices $V_i$ | Evaluation Set | SCM Evaluation Result | Membership $r_{ij}$ | | | |
|---|---|---|---|---|---|---|
| | | | $r_{1j}$ | $r_{2j}$ | $r_{3j}$ | $r_{4j}$ |
| $V_1$ | excellent | 0.85 | 0.8 | 0.7 | 1 | 0.9 |
| | good | 0.075 | 0.1 | 0.2 | 0 | 0 |
| | moderate | 0.075 | 0.1 | 0.1 | 0 | 0 |
| $V_2$ | excellent | 0.875 | 1 | 0.8 | 0.8 | 0.9 |
| | good | 0.075 | 0 | 1 | 0.2 | 0 |
| | moderate | 0.05 | 0 | 0.1 | 0 | 0.1 |
| $V_3$ | excellent | 0.775 | 0.7 | 0.7 | 0.8 | 0.9 |
| | good | 0.075 | 0.1 | 0.2 | 0.1 | 0.1 |
| | moderate | 0.1 | 0.2 | 0.1 | 0.1 | 0 |
| $V_4$ | excellent | 0.833 | 1 | 0.8 | 0.8 | 0.9 |
| | good | 0.1 | 0.2 | 0.1 | 0 | - |
| | moderate | 0.033 | 0 | 0 | 0.1 | - |

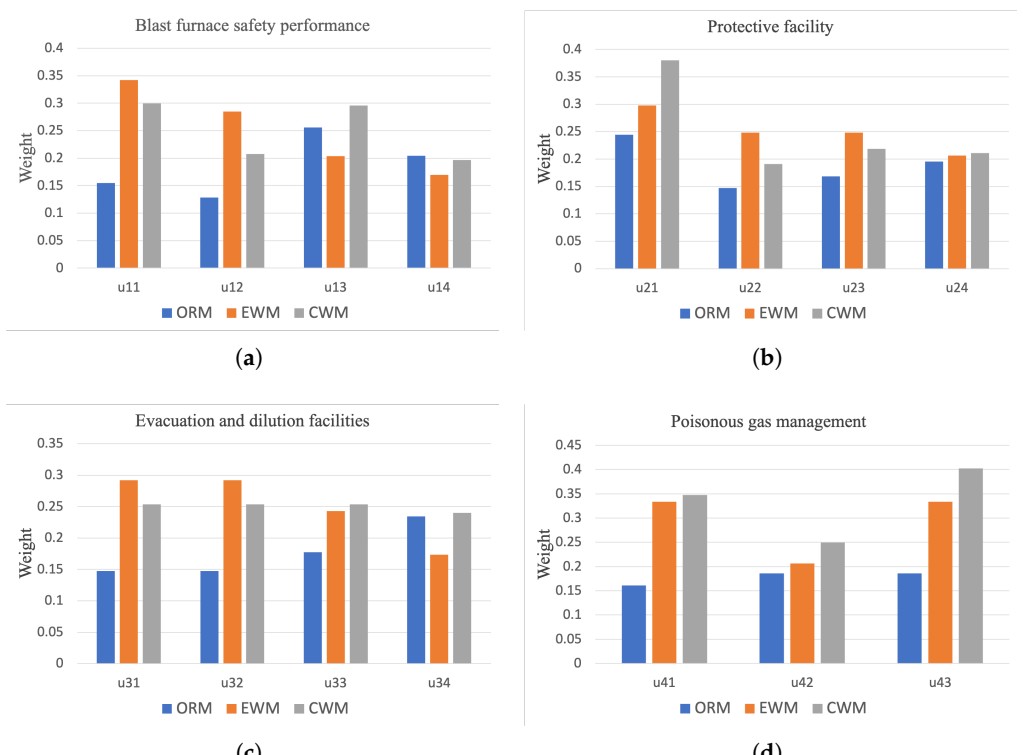

**Figure 7.** Comparison of ORM, EWM, and CWM weights for: (**a**) blast furnace safety performance; (**b**) the protective facility; (**c**) evacuation and dilution facilities; and (**d**) poisonous gas management.

The weights calculated by CWM avoided the lack of accuracy that might occur with ORM or EWM weights. For example, the weight of $u_{21}$ in the protective facility category obtained by CWM was bigger than ORM and EWM. Actually, $u_{21}$ is the most important index in the protective facility category, and the membership value $r_{21}$ is 1, which indicates that all of the experts confirmed that the safety level of $u_{21}$ was excellent. Therefore, the CWM result precisely reflected the actual situation.

*4.3. Evaluation Result Analysis of First-Level Indices*

For the blast furnace safety performance index, the evaluation vector $F_1$ was (0.8581, 0.0715, 0.0704, 0, 0). According to the principle of the maximum membership [35], the evaluation result of this index is "excellent". At the same time, according to the evaluation sets, the value of "excellent" was much higher than other evaluation sets, and there was not much difference between the values of "good" and "moderate". Thus, we can come to the conclusion that the blast furnace safety performance was stable and safe, under the current circumstances.

For the protective facility index, the evaluation vector $F_2$ was (0.8971, 0.0628, 0.0402, 0, 0). The evaluation result of this index is "excellent," with a value of 0.8971, which is the highest in this evaluation set. The sum of the values of "good" and "moderate" occupied nearly 0.1, such that the protective facilities in the plant could be considered to satisfy the overall safety demands regarding toxic gas leaks; however, there was still room for growth.

For the evacuation and dilution facilities index, the evaluation vector $F_3$ was (0.7733, 0.1253, 0.1014, 0, 0). The evaluation result of this index was "excellent". At the same time, the value of "excellent" was less than 0.8, which means that the safety status of evacuation and dilution facilities was not optimal. As a result, it can be concluded that the overall evaluation result was "excellent", but vulnerable factors in the evacuation and dilution facilities needed to be checked.

For the poisonous gas management index, the evaluation vector $F_4$ was (0.8652, 0.0945, 0.0403, 0, 0). The evaluation result of this index is "excellent". Consequently, the management of poisonous gas was proper, and the risk of toxic gas leaks caused by lack of management was relatively low.

The overall evaluation result of the toxic gas leak risk in this steel plant was "excellent", corresponding to the actual risk level of the steel plant. The results illustrate that the overall risk level of gas leaks in the considered steel plant was well under control. The factors of gas leak risk remained stable and safe, and the probability of gas leakage accidents occurring was relatively low. Meanwhile, we can see from Table 5 that the safety level of evacuation and dilution facilities was relatively low compared to the other indices, and the safety levels associated with the corrosion degree of the gas tank and CO monitoring alarm devices were relatively high. Therefore, attention should be focused on the fragile factors in the chain, according to the RAS results.

In steel plants, one of the most basic safety management methods is the safety checklist method (SCM), which is commonly adopted in industrial safety management. The steel plant in this study adopted risk classification and control, with the classification of the risk based on a safety management method. The calculation result for the safety checklist method for each first-level indicator is given as the average value of $r_{i1}$ of its secondary indices, as shown in Table 6.

In order to validate the RAS evaluation result, the RAS and SCM evaluation results were compared, as shown in Figure 8. According to Figure 8, there was little difference between the evaluation results. The excellent values of $V_2$ and $V_4$ obtained through the RAS were slightly bigger than those of SCM. The reason for this is that the importance and numerical authenticity of each index are not considered in the evaluation process for SCM; additionally, the actual situations of $V_2$ and $V_4$ in the steel plant are well-managed, and the risk of a gas leak is relatively low.

Moreover, as a quantitative analysis method, the difference of RAS from the existing risk analysis methods for steel plants is that it can accurately feedback the safety status of each first-level indicator; that is, if the risk of toxic gas leakage of this indicator is evaluated as "excellent", then its unsafe possibility and evaluation score will still be reported to the safety management department for reference. As shown in Figure 8b, the value of $V_3$ calculated by RAS was evidently bigger than that found with SCM, which indicates that the gas leak risk of evacuation and dilution facilities evaluated by RAS was not at the "excellent" status; as such, the safety management department could take targeted measures to improve the safety level of the steel plant with respect to the toxic gas leak risk.

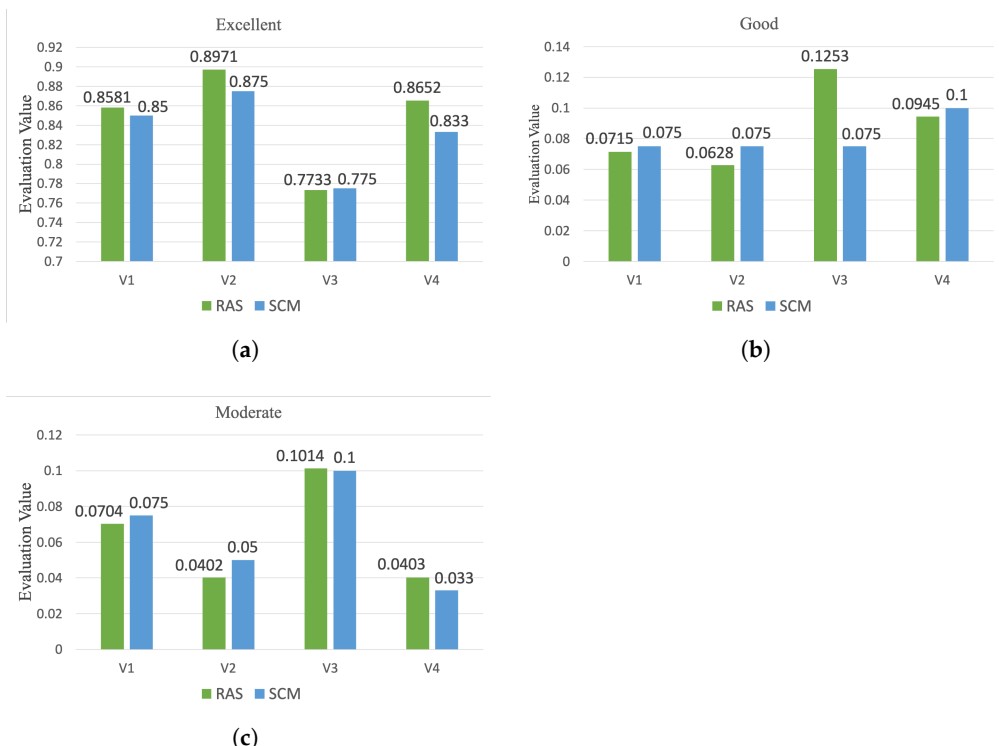

**Figure 8.** Evaluation analysis of the RAS and SCM in terms of the results for: (**a**) excellent; (**b**) good; and (**c**) moderate.

### 4.4. Application Analysis of Risk Assessment System

In order to verify the efficiency of the RAS in the application process, the time used by the RAS was recorded in the risk evaluation of toxic gas leakages, and it was compared with the time used by the existing method in the steel plant, as shown in Table 7.

**Table 7.** Time cost of the risk assessment system and safety checklist method.

| Method | Time Cost (s) | | | | | Time Reduced (%) |
|---|---|---|---|---|---|---|
| | **On-Site Scoring** | **Data Collection** | **Calculation** | **Evaluation** | **Total** | |
| RAS | 3765 | 0.08 | 0.2 | 0.1 | 3765.38 | 40.00 |
| SCM | 3524 | 612 | 1843 | 297 | 6276 | - |

The RAS takes a longer time for experts to score than the SCM because experts need to log in to the system, enter the scoring results, and submit them, while the safety checklist method only needs to record the results on the checklist when scoring. Generally speaking, the number of personnel required in the expert scoring process is similar to whether RAS or SCM is needed because the risk indicators of toxic gas leakage in steel manufacturing plants are spread at various points. A sufficient number of expert scores can ensure the accuracy of the evaluation results. In the RAS, the time cost for scoring, calculation, and risk assessment are much less than in the SCM. The reason is that the calculation can be done quickly on the server. The advantage of the RAS is that it can accurately evaluate the risk of toxic gas leakage in steel plants and then present the risk of each indicator through objective calculation in a brief period, which can provide a reference for safety management departments to respond more quickly.

RAS completed the risk assessment process in 3765.38 seconds, and the time cost of RAS was nearly half of the existing method adopted in the steel plant. On this basis, the risk analysis and evaluation results of toxic gas leakage through the RAS can clarify the

risk level of each second-level index, which is conducive to optimizing the distribution of safety management resources, reducing safety management costs, and improving safety management efficiency.

## 5. Conclusions

Toxic gas leaks and associated accidents in steel manufacturing plants can threaten the safety of workers and staff. As such, evaluating the toxic gas leakage risk in an effective and efficient way is becoming necessary for metallurgical steel plants in order to control the rising trend of toxic gas leakage accidents.

In this paper, to address the problem of inefficiency and lack of accuracy in evaluating toxic gas leak risk factors, a risk assessment system (RAS) was developed. The proposed RAS was applied in a steel plant in order to validate its efficiency and accuracy in evaluating the risk of toxic gas leaks. Fifteen secondary indices were proposed to construct the RAS index system.

The RAS weight analysis proved that CWM could effectively avoid the deviations in the evaluation that may be caused by the dependence on the order in ORM or the ignorance of risk in EWM. The proposed CWM, combining ORM and EWM, can well-reflect the weight of secondary indices in a relatively precise way. Additionally, the weight information provided by the RAS could indicate the difference in importance of secondary indices, which can direct the safety management department to focus on crucial factors in preventing gas leaks.

The results of first-level indexes are 0.8581, 0.8971, 0.7733 and 0.8652, respectively. The risk analysis of the first-level indices demonstrated that all of the indices were rated as "excellent", and the risk of gas leak was relatively low. Compared with other evaluation results of first-level indices, the "excellent" value of evacuation and dilution facilities was the only index which obtained a value less than 0.7, indicating that the safety performance of evacuation and dilution facilities should be improved before the situation deteriorates.

For further research, the proposed RAS could be made available to other industrial plants by designing suitable index systems for different hazardous factors. Simultaneously, the CWM may be modified to improve its adaptability, thus improving the performance of RAS in different conditions.

Forty percent of time could be reduced by adopting the RAS in evaluating the risk of toxic gas leaks in the steel manufacturing plant. For further research, the proposed RAS could be made available to other industrial plants by designing suitable index systems for different hazardous factors. Simultaneously, the CWM may be modified to improve its adaptability, thus improving the performance of the RAS in different conditions.

**Author Contributions:** Conceptualization, K.K.; methodology, K.K.; software, K.K. and C.M.; validation, C.M.; formal analysis, C.M.; investigation, Z.W.; resources, N.G.; writing—original draft preparation, N.G.; visualization, Z.W.; supervision, N.G.; project administration, N.G.; funding acquisition, N.G. All authors have read and agreed to the published version of the manuscript.

**Funding:** This research was funded by the Fundamental Research Fund for the Central Universities grant number FRF-BD-19-019A.

**Institutional Review Board Statement:** Not applicable.

**Informed Consent Statement:** Not applicable.

**Data Availability Statement:** The original coding data of the RAS could be provided, if required.

**Acknowledgments:** This work is supported by the Metal Smelting Major Accident Prevention and Control Technology Support Base.

**Conflicts of Interest:** The authors declare no conflict of interest.

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
