# Peer review of "A Risk Assessment System of Toxic Gas Leakages in Metallurgy Based on Fuzzy Evaluation with Comprehensive Weighting"

_applsci, doi:10.3390/app12125948_

Round 1
Reviewer 1 Report
Scientific articles should not use the form "we". The impersonal form is used.
..., we further evaluate the toxic gas...
..., we can calculate the...
we obtain F2=(0.8971, 0.0628, 0.0402, 0, 0)
we can conclude that the overall evaluation...
F2=(0.8971, 0.0628, 0.0402, 0, 0) - Incorrect mathematical notation.
Author Response
Thank you for your comments on our manuscript entitled as ‘A risk assessment system of toxic gas leakage in metallurgy based on fuzzy evaluation with comprehensive weighting’ (Manuscript ID: applsci-1723088). These comments are all valuable and very helpful for revising and improving our paper, as well as the important guiding to our researches. We have studied the comments carefully and have made corrections, which we hope meet with approval.
The point-by-point responses to your comments are presented in the attachment.
We hope with these revisions, our manuscript can be accepted to publish in the Applied Sciences.

Reviewer 2 Report
This article presents an analysis of the multi-criteria task that occurs in metallurgy in the area of elimination of toxic gas leaks. The article deals with practical solutions that are used in industry.
The results of the assessment are in line with the expectations of the steelworks. Therefore, it is necessary to consider the implementation of software that could accelerate the performance of such analyzes and at the same time be made available to other industrial plants after the solution is patented.
I do not have any more comments on the submitted article.
Author Response

(The authors gave the same response as above.)

Reviewer 3 Report
In order to increase the quality of the current article, the following comments should be addressed:
1. Abstract of this paper needs to improve in view of existing problems and the solution adopted by the new methodology.
2. Redraft the introduction including background, challenges, a literature survey of recent works, research scopes, motivation, objectives, contribution, and organization of the paper.
3. Reorganize the paper to maintain the workflow of the entire paper, especially during the transition between two sections/subsections.
4. Please add more simulation results and compare these results with existing approaches to show the superiority of the proposed approach.
5. Finally, there are lots of grammatical errors and typos. Please thoroughly check the whole paper to correct all of these errors.
Author Response

(The authors gave the same response as above.)

Reviewer 4 Report
The article ‘A risk assessment system of toxic gas leakage in metallurgy based on fuzzy evaluation with comprehensive weighting’, however, there are many major and minor aspects authors might need to concentrate before the article is considered for the publication in this journal. Following is the feedback for the authors:
- The first line of the abstract is not well defined as problem statement. Please critically revised it as why there is need of this study. Reflect the problem as emerging need in the current scenario.
- The abstract is very general, it should have some very important findings statistics and particular conditions etc. Authors should add some numeric / figured values for better understanding and good readership.
- Abstract should be revised thoroughly, in its current form it gives very basic information, which does not reflect the critical research paper structure.
- The introduction section only perspective of china, authors should add the state of the art literature comparative studies from different parts of the world.
- Most of the references are from chines authors, which gives one sided picture of the story. It is encouraged to add latest references relevant to the study.
- Figure 1 ‘’ Crude steel production of China from 2013 to 2020’’ provide a general information. Please note that this is a research paper. Normally in the intro section, figure / table is not recommended unless very critical view. This is very general data and should be deleted.
- Method section contain a lot of references, which should be minimized and information should be discussed in the introduction section. In the method section authors should discuss their own methodology.
- Table 1 in fact give no meaning and very basic one liner information. It should be rather inside the text.
- Headings/ sub-headings are normally not recommended in the abbreviation form. Please check thoroughly in whole paper.
- Authors have used many mathematical equations. Are these authors’ own equations or taken from literature? If so, please give references.
- Again table 2 is very general information. Tables should include some important parameters. One liner tables should be avoided as tabular form.
- Again, heading 3 include abbreviation.
- Almost all references are from chines authors, which is highly discouraged. Authors should include worldwide discussions for better readership.
- There is no significant literature. This is not a good study unless addition of some relevant literature.
- Methodology section is too long. Authors needs to short it. It is not mandatory to include every information in the methodology section.
- Results section is very week, authors are required to re-work on results section for the critical results and their analyses. It is very difficult to give a lot of major comments on the results section. It is better to rethink about the critical discussion and graphs / results display. In the present form, it is not good R&A section. In fact authors have written ‘’application section, instead of results section. Authors should include proper results section as well.
- Conclusion section is very general, authors should re-write this section only focusing on the outcome of the present results.
Author Response

(The authors gave the same response as above.)

Round 2
Reviewer 4 Report
Authors have tried to address the feedback, however the article can be still improved for the technical content. Article can be considered accepted after going through the minor checks by the authors.